# Mechanical Properties of New Generations of Monolithic, Multi-Layered Zirconia

**DOI:** 10.3390/ma16010276

**Published:** 2022-12-28

**Authors:** Maria Bruhnke, Yasmin Awwad, Wolf-Dieter Müller, Florian Beuer, Franziska Schmidt

**Affiliations:** Department of Prosthodontics, Geriatric Dentistry and Craniomandibular Disorders, Charité—Univeritätsmedizin Berlin, Corporate Member of Freie Universität Berlin, Humboldt—Universität zu Berlin and Berlin Institute of Health, Aßmannshauser Straße 4-6, 14197 Berlin, Germany

**Keywords:** zirconia, mechanical properties, multi-layer materials, monolithic ceramic, dental ceramic

## Abstract

New monolithic multi-layered zirconia restorations are gaining popularity due to their excellent aesthetic properties. However, current knowledge of these newest multi-layer ceramics in terms of mechanical properties is scarce. Three monolithic, multi-layered zirconia materials (Katana, Kuraray Noritake, Japan) were selected for comparison: High Translucent Multi-layered zirconia (HTML), Super Translucent Multi-layered zirconia (STML) and Ultra Translucent Multi-layered zirconia (UTML). Fifteen specimens per group were cut from pre-sintered blocs in each of the four layers (L1, L2, L3, L4) and in different thicknesses (0.4 mm, 0.8 mm and 1.2 mm). Critical fracture load (*Fcf*) was recorded in 3-point-bending. Flexural strength (*σ*) in MPa, Vickers hardness (*HV*) in N/mm^2^, fracture toughness (*K_Ic_*) in MPa*m1/2, Weibull Modulus (*m*) and characteristic Weibull strength (*σw*) in MPa were assessed. Statistical analysis was performed using ANOVA analysis. *FS* and *K_Ic_* were significantly higher (*p* < 0.05) for Katana™ HTML (652.85 ± 143.76–887.64 ± 118.95/4.25 ± 0.43–5.01 ± 0.81) in comparison to Katana™ STML (280.17 ± 83.41–435.95 ± 73.58/3.06 ± 0.27–3.84 ± 0.47) and UTML (258.25 ± 109.98–331.26 ± 56.86/2.35 ± 0.31–2.94 ± 0.33), with no significant differences between layers and layer thicknesses. The range of indications should be carefully considered when selecting the type of monolithic zirconia for fabrication of dental restorations, as materials widely differ in mechanical properties.

## 1. Introduction

Zirconia is gaining popularity as the restoration material of choice for dental prostheses and is a viable alternative to conventional metal–ceramic restorations. The main advantages of zirconia are its biocompatibility [1], biomechanical stability, high fracture resistance [2] and its tooth-like, esthetic appearance. Due to its excellent mechanical properties, Yttria-stabilized Zirconia (3Y-TZP) ceramic is used as a framework material [3,4]. Based on the high opacity of the material, restorations are typically veneered with a weak feldspar- or glass-ceramic material [5]. Research has shown that these bi-layered restorations are susceptible to fracture, delamination and chipping of the veneering material [6,7,8], accounting for an annual complication rate of 3.14% [9]. The search for a monolithic material that overcomes these major technical complications and disadvantages has become a driving force for material development [10]. First generation zirconia materials consist of a tetragonal stabilized polycrystalline zirconia (TZP: tetragonal zirconia polycrystal) doped with 3 mol% yttrium oxide (3Y-TZP). The high opacity of 3Y-TZP forced the development of more translucent zirconia materials [11]. Translucency is usually achieved with a higher percentage of the cubic phase, which may be generated by increasing sintering temperatures and/or altering the yttria content [12]. With higher yttrium oxide doping, the proportion of the cubic phase is increased by approximately 25 percent for 4Y-TZP and up to approximately 53 percent for 5Y-TZP, allowing for a higher degree of translucency [13]. The addition of dopants and coloring liquids will change the color and increase opacity. However, this increase in cubic phase content goes along with a decrease in mechanical strength, as the exceptional flexural strength and fracture toughness are due to the transformation toughening mechanism, which only occurs in the metastable tetragonal phase. An increase in yttria content will lead to an increase in translucency and a decrease in mechanical strength.

Mechanical properties of ceramics are typically assessed by different material-dependent values such as flexural strength, Weibull modulus, fracture toughness and Vickers hardness [14]. In a flexural strength experimental setup, test specimens are loaded under bending stress with increasing forces until fracture. Flexural strength is determined by the size and number of defects in the material’s microstructure, and is therefore a material-dependent value. It shows an asymmetrical strength distribution described by Weibull statistics and the Weibull modulus, which is a measure for the scatter of strength data related to the defect size and defect distribution [15]. The lower the scatter, the higher the Weibull modulus. Fracture toughness describes the ability of a ceramic material to prevent crack propagation.

More recently, new monolithic multi-layer ceramic materials have been developed with different color and translucency gradients throughout the layers in order to improve semblance to natural tooth structure: while the incisal part of teeth is more translucent, the cervical area appears opaquer [16]. Zirconia multi-layer discs are pre-sintered, consisting of four different layers with varying degrees of translucency and color intensity: the outer enamel layer, two transitions layers, and an inner body layer (manufacturer’s information). The degree of translucency decreases from the outer to the inner layer, while color intensity increases, respectively. However, current knowledge of these newest multi-layer ceramics in terms of the mechanical properties is scarce [17]. As the zirconia materials are indicated in similar clinical situations as a metal based one, but exhibit different material compositions and properties, the subject of this investigation was to assess the mechanical properties of these materials.

Thus, the object of the study was to investigate whether there is a difference in mechanical properties in terms of flexural strength, Vickers hardness, Weibull modulus and fracture toughness of three different Katana™ Zirconia Multi Layered blanks: Katana Zirconia High Translucent Multi-layered (HTML), Super Translucent Multi-layered (STML) and Ultra Translucent Multi-layered (UTML). The second aim of this study was to investigate whether there is a difference between the layers within a blank. Furthermore, another advantage of these pre-layered blanks is the possibility of reducing the layer thickness due to the lack of the veneering material [17,18]. Therefore, the third objective of the study was to investigate whether a reduction in layer thickness influences mechanical properties of the tested materials. The research hypotheses are: there are no differences in mechanical properties between the three different materials under investigation, between different layers within a blank, and different layer thicknesses.

## 2. Materials and Methods

### 2.1. Materials

Three commercially available pre-sintered Katana™ Zirconia Multi Layered materials were selected in this comparative study: Katana™ Zirconia Multi Layered (ML), Katana™ Zirconia Super Translucent Multi Layered (STML) and Katana™ Zirconia Ultra Translucent Multi Layered (UTML). Table 1 provides an overview of the tested ceramic materials with their chemical compositions according to the manufacturer’s information. Sample size was calculated according to a pilot study, considering α = 0.05 and β = 0.80 with n = 15 per group.

### 2.2. Specimen Preparation

Preparation of specimens for the determination of mechanical properties was performed in accordance with the DIN standard ISO 6872:2015. Bar-shaped specimens were cut from pre-sintered ceramic blocs using a band saw (Exakt Advanced Technologies GmbH, Norderstedt, Germany) in four different layers (L1, L2, L3, L4) horizontally (Figure 1). Specimens were sintered in a furnace (Sinterofen Denta-Star P1 plus, Thermo-Star GmbH, Aachen, Germany) according to the manufacturer’s instructions with the following parameters: 7 h from room temperature to 1500 °C at 10 °C/min for HTML, and to 1550 °C for STML and UTML, holding time of two hours at firing temperature and cooling to room temperature at 10 °C/min. After sintering, the final specimens for hardness and flexural strength were cut from the sintered blocks with a diamond precision saw (Isomet, Low Speed Saw, Isomet Precision Saw, Buehler, IL, USA.) under constant water cooling. They were cut into samples with varying thickness of 0.4 mm, 0.8 mm and 1.2 mm, resulting in bar-shaped specimens with final dimensions of 12 mm × 2.4 mm × (0.4/0.8/1.2 mm). Specimens were sequentially polished from coarse to fine with silicone carbide coated abrasive paper (Carbi Met, P800, P1200, P2500, Microcut P4000, Buehler, Lake Bluff, IL, USA) under constant water cooling (Poliermaschine LaboPol-25, Struers GmbH, Willich, Germany). The surface of the specimen was cleaned, degreased and dried with acetone prior to testing.

### 2.3. Mechanical Properties Assessment

Vickers hardness (*HV*) was measured with a microhardness tester (Q10M, Qness GmbH, Golling, Austria). Specimens were mechanically loaded with a defined force F of 49.5 N (*HV*) perpendicular to the surface using a Vickers diamond indenter in the shape of a pyramid with an inclination angle of 136° at the tip. A minimum of six indentations were performed per specimen. The indentation was measured by optical evaluation with a microscope with a magnification of 40 and 65 times and an integrated 18 Megapixel camera, and electronically assessed with the aid of an image-editing program (Software Qpix T2, Qness GmbH, Golling, Austria). Vickers hardness *HV* in MPa was calculated according to the following formula:(1)$$HV=\frac{0.189\cdot F}{d^{2}}$$
where 0.189 is a constant value, *F* = test force in N, and *d* = average of the indentation diagonal in mm. For calculation of the fracture toughness (*K_Ic_*) in MPa√m by indentation fracture (IF) method according to Anstis et al. [19], the following formula was used:(2)$$K_{Ic}=0.032\cdot a^{1/2}\cdot Hv\cdot {\left(\frac{E}{Hv}\right)}^{1/2}{\left(\frac{c}{a}\right)}^{-3/2}$$
where hardness was calculated according to Equation (3):(3)$$Hv=\frac{F}{2a^{2}}$$
with *a* = ½ *d* (indentation diagonal length) in mm, *c* = ½ of crack length in mm, *E* = Elastic modulus of Zirconia (as supplied by the manufacturer, see Table 1).

Flexural strength *σ* (FS) in MPa was assessed by three point bending tests in a universal testing machine (Zwick Z010, Zwick Roell, Ulm, Germany) at 1 mm/min crosshead speed. The critical fracture load (F_max_) in Newton (N) until crack formation was measured. FS was calculated as follows:(4)$$\sigma =\frac{3LF}{2wb^{2}}$$
where *L* = distance between the outer support in mm, *F* = maximum fracture load in *N*, *w* = width of specimen in mm and *b* = thickness of specimen in mm.

Weibull analysis was employed for reliability evaluation of the flexural strength of the different Zirconia materials. For each material FS values (*n* = 20) were obtained and sorted in ascending order. Weibull modulus (m) and the characteristic strength in MPa (*σ_w_*) were obtained from the Weibull diagram, applying the Equation (5) with *p*—fracture probability, *σ*—flexural strength.
(5)$$p\left(FS\right)=1-exp{\left(\frac{\sigma }{\sigma^{w}}\right)}^{m}$$

### 2.4. Statistics

Statistical analysis was performed with OriginLab (Origin Pro 2020 b Software GmbH, Erkrath, Germany). After assessment of the normal distribution using the Kolmogorov–Smirnov test the data were analyzed by one-way analysis of variance (one-way ANOVA) with post hoc Scheffé and Bonferroni for comparison of mean values. For all tests the significance level was set at *p* < 0.05.

## 3. Results

The data were normally distributed, therefore parametric tests for statistical comparison were applied. Descriptive statistics with means and standard deviations (SD) for flexural strength (*σ*) in MPa, Vickers hardness (*HV*) in MPa and fracture toughness (*K_Ic_*) in MPa√m are presented in Table 2. Flexural strength values for Katana™ UTML, STML and HTML differed significantly (*p* < 0.05), in HTML ranging from 652.85 ± 143.76 to 887.64 ± 118.95, in STML from 280.17 ± 83.41 to 435.95 ± 73.58 and in UTML from 228.67 ± 39.34 to 331.26 ± 56.86. For flexural strength *σ* within one material there were no significant differences between layers L1, L2, L3 and L4, and between different layer thicknesses 0.4 mm, 0.8 mm and 1.2 mm (*p* > 0.05) within each material. However, when combining all layers in the same layer thickness within one material, a significant difference (*p* < 0.05) could be found between flexural strength values of samples with 0.4 mm layer thickness and with higher layer thickness of 0.8 and/or 1.2 mm layer thickness. Figure 2 shows the distribution of the flexural strength between groups. In HTML, the mean value of flexural strength of 0.4 mm layer thickness over all layers was 733.51 ± 135.70 MPa, and in layer thickness 0.8 mm the mean value was 720.68 ± 99.21 MPa. Both mean values do not differ significantly but show significant difference to the mean value of flexural strength for samples with 1.2 mm (794.78 ± 138.58 MPa). In STML the mean values ranged from 307.31 ± 111.03 MPa for 0.4 mm layer thickness to 388.71 ± 115.00 MPa for 0.8 mm layer thickness and 379.48 ± 76.30 MPa for 1.2 mm layer thickness. The mean value for 0.4 mm layer thickness shows significant difference from the values from 0.8 mm and 1.2 mm. In UTML, the mean values ranged from 270.99 ± 82.44 MPa for 0.4 mm to 283.78 ± 58.44 MPa for 0.8 mm layer thickness to 300.67 ± 43.78 MPa for 1.2 mm layer thickness. There are significant differences between mean values of 0.4 mm and 1.2 mm.

Vickers hardness *HV* was 1354.5 ± 26.94–1396 ± 31.63 MPa for HTML, 1360.83 ± 35.14–1415.67 ± 28.61 for STML and 1333.83 ± 18.49–1441.5 ± 62.79 MPa for UTML. Means and standard deviations for *HV* indicate no significant differences between layers L1, L2, L3 and L4, between layer thicknesses 0.4 mm, 0.8 mm and 1.2 mm and between tested materials Katana™ UTML, STML and HTML (*p* > 0.05).

Fracture toughness values varied between different Katana™ materials, with lowest values in UTML and highest values in HTML (Figure 3). The mean values were significantly different between tested materials (*p* < 0.05): HTML (4.25 ± 0.43–5.01 ± 0.81), STML (3.06 ± 0.27–3.84 ± 0.47), UTML (2.35 ± 0.31–2.94 ± 0.33). Within each material there were no significant differences for fracture toughness found between different layers L1, L2, L3 and L4 and layer thicknesses 0.4 mm, 0.8 mm and 1.2 mm (*p* > 0.05).

Results of the Weibull moduli and characteristic strength are given in Table 3. Weibull moduli and characteristic strength were determined for each material, separated by layer and layer thickness, and they were also determined for each material and layer thickness, with all layers L1–L4 combined. Between all the materials, HTML showed the highest values for characteristic Weibull strength, ranging from 705.52 (L1, 1.2 mm layer thickness) to 938.73 MPa (L4, 1.2 mm layer thickness). UTML showed the lowest values for characteristic Weibull strength, ranging from 290.00 MPa (L4, 0.4 mm layer thickness) to 329.54 MPa (L1, 0.4 mm layer thickness). STML values ranged from 311.71 MPa (L4, 0.4 mm layer thickness) to 467.00 MPa (L2, 0.8 mm layer thickness). There was no clear trend regarding the characteristic Weibull strength between different layers or layer thickness within one material group.

Weibull moduli in HTML ranged from 5.27 (L1, 0.4 mm layer thickness) to 11.95 (L2, 0.8 mm layer thickness). Weibull moduli in STML ranged from 2.12 (L3, 0.4 mm layer thickness) to 7.77 (L1, 1.2 mm layer thickness). Weibull moduli in UTML ranged from 2.84 (L4, 0.4 mm layer thickness) to 14.90 (L2, 1.2 mm layer thickness). A low Weibull modulus correlates to a large variation in strength values within one group. Therefore, a higher Weibull modulus correlates to a low variation in values and a higher reliability. Within each material, Weibull moduli for 0.4 mm layer thickness were lowest, showing a low reliability for this layer thickness. Higher layer thicknesses of 0.8 and 1.2 mm led to higher Weibull moduli and thereby to better reliability of the materials’ strength. This is also visible when the Weibull modulus is calculated of all layers L1 to L4 together (Figure 4). Within each material group, it is visible that the slope is steeper when strength was measured in higher layer thickness.

## 4. Discussion

This comparative in vitro study investigated the mechanical properties of three different multi-layer zirconia ceramics: Katana™ HTML, Katana™ UTML and Katana™ STML. Moreover, this is the first study reporting on the mechanical properties of different layers within one specific blank of Katana™.

The highest flexural strength was recorded for Katana™ HTML, followed by STML and UTML, as expected. The present results indicate an inverse correlation between translucency and flexural strength, and they are in accordance with other laboratory studies: a higher degree of translucency is associated with lower flexural strength [20,21] and the presence of cubic ZrO_2_ crystals [21]. A different microstructural composition of Katana™ STML, UTML and HTML accounts for the different mechanical properties. It was reported by Harada et al. that Katana™ HTML exhibits 5.6 wt%, Katana™ STML 8.15 wt% and Katana™ UTML 9.32 wt% of Y_2_O_3_ content and an increased amount of cubic phase (Table 1). In addition, STML and UMTL have a higher sintering temperature, which affects the grain size and the translucency [10].

Flexural strength values recorded in this study for Katana™ are lower than in comparable studies [20,22]. A study by Kwon et al. [20] used a similar three-point bending test set-up, reporting flexural strength values of 1194 ± 111 MPa for Katana™ HTML and 688 ± 159 MPa for Katana™ UTML. However, specimens were cut in other dimensions of 25 mm × 4 mm × 2 mm, resulting in thicker bending samples, and therefore the data may not be directly comparable [20].

In another study investigating aging resistance, mechanical properties, and translucency of different yttria-stabilized zirconia ceramics for Katana™ UTML, flexural strength of 450 MPa is reported in a piston-on-three-balls strength test [22]. With an increasing cubic phase content, a higher degree of translucency is achieved at the expense of mechanical properties such as flexural strength and fracture toughness [22]. The only comparable study, investigating the effect of masticatory simulation and the effect of layer thicknesses varying between 0.7 mm and 1.2 mm, published flexural strength values of 538 ± 45 MPa for Katana™ HTML, 454 ± 39 MPa for Katana™ STML and 307 ± 31 MPa for Katana™ UTML studied in 0.7 mm thick layers [23]. The study established a correlation between the increase of layer thickness and the increase of flexural strength values which could not be shown in the present study [23].

Moreover, the recorded values are not consistent with manufacturer’s specifications (Katana™ UTML: 557 MPa; Katana™ STML: 748 MPa; Katana™ HTML: 1125 MPa). The wide range of published values on mechanical properties in the literature shows the difficulty of a direct comparison. Data are dependent on different variables such as experimental set-up, testing parameters and the test procedure [24]. The milling and surface preparation procedure itself have an especially significant influence on the flexural strength of tested ceramic materials [25]. It should be noted, as well, that most samples in this study were prepared from a maximum of three blanks for each material, which is limiting. There are variations in the productions process that may result in different characteristics. In addition, most laboratory furnaces are not calibrated, which may lead to variations in the applied sinter protocol as well. However, a direct comparison of tested materials within one study seems reasonable and it is in accordance with other studies: flexural strength increases from Katana™ UTML, to Katana™ STML, to Katana™ HTML [20,23]. Mechanical properties are remarkably superior for Katana™ HTML, while for Katana™ UTML and Katana™ STML this study demonstrates moderate differences.

Fracture toughness is considered as an additional and essential factor for the assessment of mechanical properties of ceramics [2]. It is an intrinsic material property that measures the resistance against crack propagation, and therefore how much stress a brittle material can absorb before the catastrophic failure [26]. Results in this study are in accordance with previous studies [22]. However, again, comparison to other studies should be carried out with caution as fracture toughness measurements are dependent on method and the type of introduced defects [26].

Fracture toughness by IF method, as was undertaken in this study, is comparably simpler in preparation than other methods, such as single edge notch beam or Chevron notch beam tests. According to Anstis et al. there is only about a 30% correlation between fracture toughness values by IF and other methods [27]. However, due to its simplicity, the IF method allows a reliable comparison of the different qualities of the tested ceramics in relation to their indication. The measured *K_Ic_* values indicate significant differences between tested materials Katana™ HTML, STML and UTML, and no significances between layers in one material. Data on Vickers hardness are consistent and in line with the scientific literature [22].

Data over all layers show that specimens in Katana™ STML and UTML groups exhibited slightly lower Weibull moduli than in HTML, indicating scattering in results attributed to physical structural properties of the ceramic material, especially the size and distribution of defects. In contrast to conventional 3Y-TZP, zirconia ceramics with higher yttria content exhibit lower tetragonal phase content and therefore less transformation toughening with simultaneous crack closure mechanisms. Data are in line with another study [28]. Within each material group the Weibull moduli clearly increased with higher sample thickness, indicating less scattering of the results, as it could be assumed that the defect distribution is dependent on the thickness. Possibly the surface defects are more prone in case of the thinner specimens. Therefore, it seems to be recommended to test specimens with a thickness of 1,2 mm. Otherwise, for the real application and recommendation of indication, it seems necessary to recognize that the thinner the dimensions of the real structures are the more critical the survival condition for such devices are.

Chemical composition, microstructure and translucency properties were assessed for Katana™ STML, UTML and HTML in another laboratory study [16]. The authors found small differing pigment and contents within layers, but no significant differences in translucency and composition. This finding is in accordance with results of the present study, as there were no significant differences in mechanical properties between layers as well.

Measurement specifications for mechanical properties for dental ceramics are suggested by the ISO-6872 standard [29]. The norm offers specific requirements for a wide range of mechanical properties, such as Weibull statistics and flexural strength values. Moreover, fracture toughness is considered an important parameter for the determination of mechanical properties. Finally, it is the combination of different mechanical properties that leads to a reasonable assessment for clinical indication.

For monolithic fixed partial dentures with ≥4 units, the norm specifies a minimum of 800 MPa for flexural strength. As shown in Figure 5, none of the tested materials fulfill these criteria, even that proposed by Filser et al. [2]. Katana™ HTML may be used for monolithic restorations with up to three units (anterior, premolar and molar region) (>500 MPa). Three-unit fixed partial dentures (anterior and premolar region) and single crowns with conventional cementation protocols may be fabricated with Katana™ HTML and STML (>300 MPa). All tested materials fulfill the requirement for veneers, inlays, onlays and single crowns in the posterior and anterior region with adhesive cementation protocols. However, an in vitro study has demonstrated a negative effect on three-dimensional external marginal gap progression after chewing simulation when comparing STML with zirconia-reinforced lithiumdisilicate crowns on endodontically treated teeth with different preparation designs [30]. The authors conclude that further studies are necessary to provide a better understanding of the biomechanical behavior of these materials.

Limitations of the present study are a lack of optical evaluation with the aid of translucency tests, microstructural analyses and material compositions in terms of yttria-content, grain size and distribution, lack of testing of aging behavior and hydrothermal degradation. These variables need further investigation. Furthermore, future research should investigate mechanical properties and fracture behavior of fully anatomical and zirconia frameworks after thermocycling, mechanical loading and aging behavior prior to clinical application.

Although new multi-layered zirconia materials have optimal esthetic qualities, results implicate high differences in flexural strength and fracture toughness between conventional and more translucent zirconia materials. Biaxial strength of the investigated new highly translucent multi-layer ceramics is even comparable to lithium-disilicate ceramics [20,22,31]. Therefore, mechanical properties of new translucent multi-layered zirconia materials should be carefully evaluated, especially in the posterior region with wide span fixed partial dentures, and zirconia-reinforced lithium silicate glass-ceramics have to be considered as alternative materials. Moreover, the term “zirconia” includes a wide range of materials with different mechanical properties that should be taken into account before clinical application.

## 5. Conclusions

Within the limitations of this in vitro study design, flexural strength and fracture toughness was significantly higher for Katana™ HTML in comparison to Katana™ STML and UTML;The correlation between fracture toughness and bending strength corresponds to that proposed by Filser et al. [2] and shows that the use of the IF technique to determine *K_Ic_* seems relatively reliable and thus suitable as a basis for the indication;The range of indications based on these data should be carefully considered when selecting the type of monolithic zirconia material for fabrication of dental restorations, as materials widely differ in mechanical properties.

## Figures and Tables

**Figure 1 materials-16-00276-f001:**
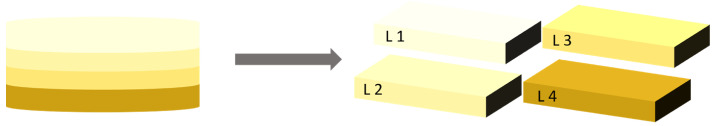
Scheme of layer structure of multilayer zirconia blanks and specimen preparation from each layer. L1 = enamel layer, L2 = first transition layer, L3 = second transition layer, L4 = dentin layer.

**Figure 2 materials-16-00276-f002:**
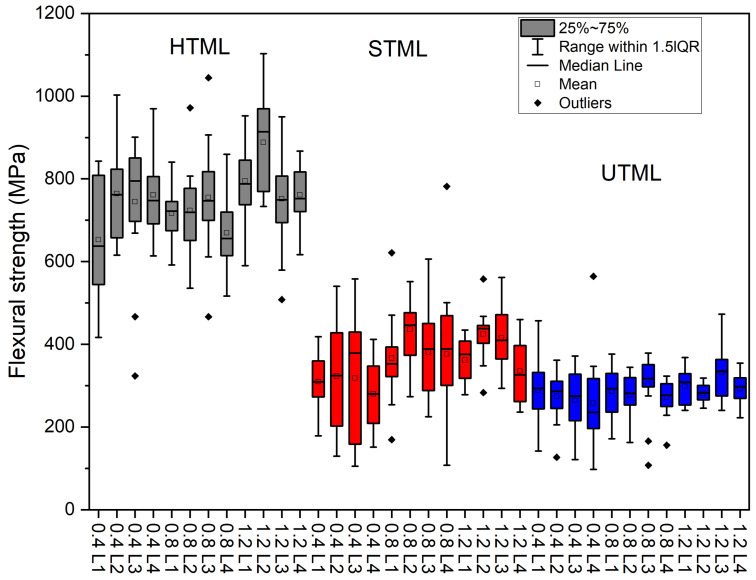
Flexural strength *σ* [MPa] for multilayer samples shown by boxplots. HTML: Katana™ Multi-layered zirconia; STML: Katana™ Super Translucent Multi-layered zirconia; STML: Katana™ Ultra Translucent Multi-layered zirconia.

**Figure 3 materials-16-00276-f003:**
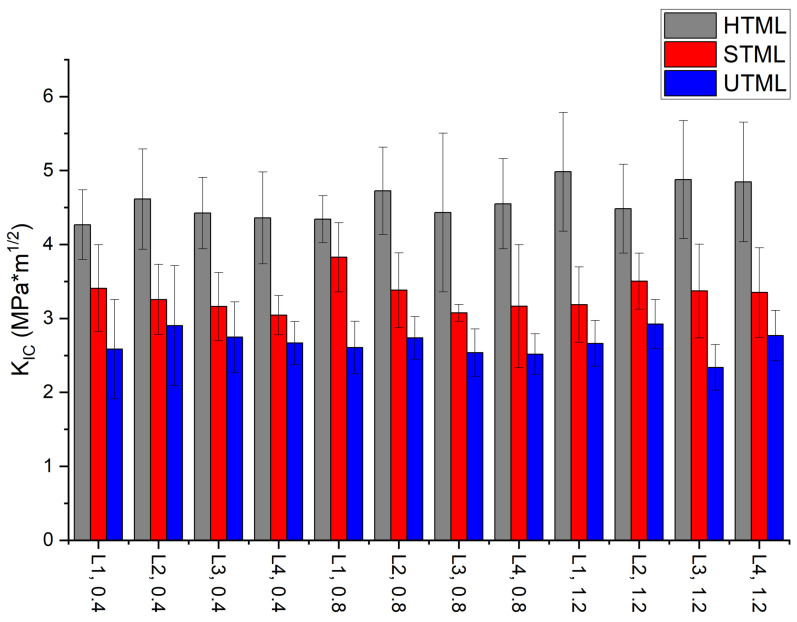
Fracture toughness *K_Ic_* ín MPa√m for multilayer samples. HTML: Katana™ High Translucent Multi-layered zirconia; STML: Katana™ Super Translucent Multi-layered zirconia; STML: Katana™ Ultra Translucent Multi-layered zirconia.

**Figure 4 materials-16-00276-f004:**
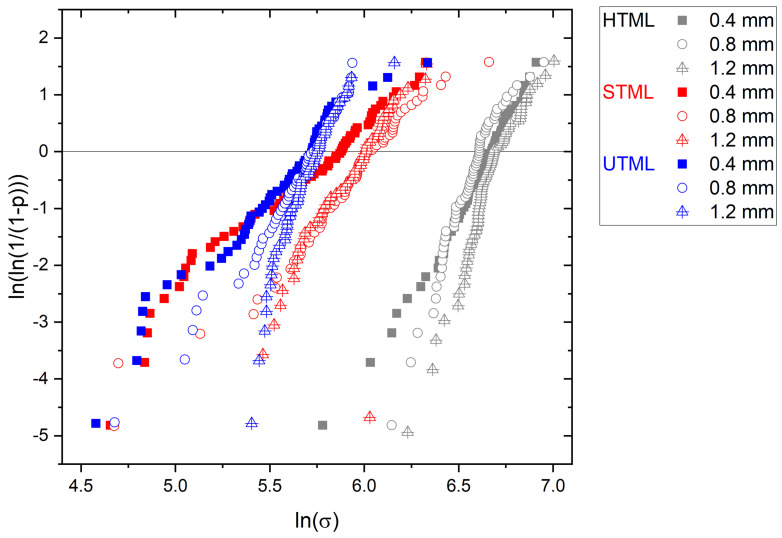
Weibull plots of the Katana materials separated according to layer thickness. HTML: Katana™ High Translucent Multi-layered zirconia; STML: Katana™ Super Translucent Multi-layered zirconia; STML: Katana™ Ultra Translucent Multi-layered zirconia.

**Figure 5 materials-16-00276-f005:**
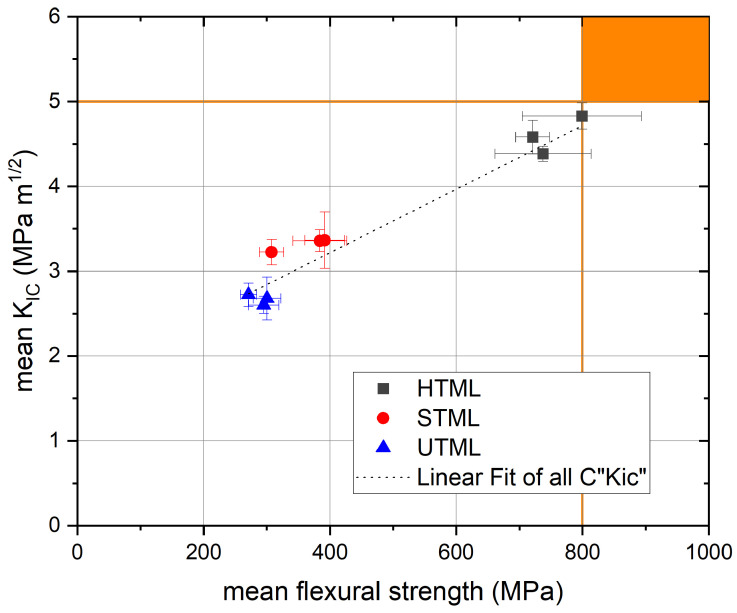
*K_IC_* vs. Bending strength (*σ_b_*) for Katana HTML–STML–UTML multilayer ceramics based on Filser et al. [2]. The right upper corner in orange presents the range for unlimited indication.

**Table 1 materials-16-00276-t001:** Overview of tested Zirconia materials, LOT numbers and different properties according to the manufacturer’s information (Katana, Kuraray Noritake, Japan).

Zirconia Material	Wt.-% Cubic Phase	Grain Size in µm	Flexural Strength (MPa)	Translucency ^1^ (%)	Elastic Modulus (GPa)
KATANA HTML [LOT: DQVMR]	<50	0.63 (±0.03)	1125	31	214
KATANA STML [LOT: DRYQY]	~65	2.81 (±0.17)	748	38	217
KATANA UTML [LOT: DQUGC]	~75	4.05 (±0.85)	557	43	217

^1^ Translucency is defined as light transmission: 100% (transparent) and 0% (opaque).

**Table 2 materials-16-00276-t002:** Descriptive non-parametric statistics (Mean ± SD) for flexural strength *σ*, Vickers hardness *HV* and fracture toughness *K_Ic_* for all tested groups.

Flexural Strength *σ* [MPa]						
		L1 (Mean ± SD)	L2 (Mean ± SD)	L3 (Mean ± SD)	L4 (Mean ± SD)	All Layers (Mean ± SD)
HTML	0.4 mm	652.85 ± 143.76 C	764.15 ± 119.97 ABC	744.55 ± 143.76 BC	761.38 ± 107.06 ABC	733.51 ± 136.81 D
	0.8 mm	716.71 ± 69.62 BC	723.55 ± 100.23 BC	754.83 ± 133.22 ABC	669.54 ± 86.11 BC	715.27 ± 103.24 D
	1.2 mm	794.72 ± 93.37 AB	887.64 ± 118.95 A	751.84 ± 105.96 BC	761.14 ± 67.35 ABC	792.79 ± 108.23 E
STML	0.4 mm	309.72 ± 66.38 AB	322.90 ± 136.42 AB	317.60 ± 146.17 AB	280.17 ± 83.41 B	307.31 ± 111.03 D
	0.8 mm	362.92 ± 57.12 AB	435.95 ± 73.58 A	380.22 ± 115.32 AB	387.48 ± 75.80 AB	388.71 ± 115.00 E
	1.2 mm	361.34 ± 53.50 AB	424.50 ± 76.68 AB	414.84 ± 76.66 A	335.25 ± 72.55 AB	379.48 ± 76.30 E
UTML	0.4 mm	287.97 ± 84.11 A	273.84 ± 57.98 A	263.89 ± 77.80 A	258.25 ± 109.98 A	270.99 ± 82.44 D
	0.8 mm	285.94 ± 63.08 A	284.41 ± 38.56 A	331.09 ± 35.11 A	279.52 ± 31.24 A	283.78 ± 58.44 DE
	1.2 mm	297.95 ± 37.68 A	280.33 ± 22.45 A	331.26 ± 56.86 A	292.35 ± 38.53 A	300.67 ± 43.78 E
Vickers Hardness *HV* [MPa]						
		L1 (Mean ± SD)	L2 (Mean ± SD)	L3 (Mean ± SD)	L4 (Mean ± SD)	All Layers (Mean ± SD)
HTML	0.4 mm	1379.83 ± 9.41 A	1354.5 ± 26.94 A	1387.17 ± 15.25 A	1396 ± 31.63 A	1379.38 ± 17.85
	0.8 mm	1374.67 ± 18.13 A	1379.5 ± 19.15 A	1386.83 ± 22.40 A	1379.17 ± 24.88 A	1380.04 ± 5.03
	1.2 mm	1382.4 ± 12.25 A	1365.5 ± 19.34 A	1379.33 ± 18.98 A	1375.00 ± 20.72 A	1375.56 ± 7.36
STML	0.4 mm	1373.5 ± 15.71 AB	1415.67 ± 28.61 A	1378.83 ± 6.43 AB	1407.67 ± 30.81 A	1393.92 ± 20.87
	0.8 mm	1384.17 ± 22.95 AB	1403.67 ± 55.38 A	1371.5 ± 22.58 AB	1379.17 ± 24.31 AB	1384.63 ± 13.72
	1.2 mm	1364 ± 18.67 AB	1367.5 ± 19.81 AB	1343.83 ± 22.59 B	1360.83 ± 35.14 AB	1359.04 ± 10.50
UTML	0.4 mm	1370.33 ± 15.97 AB	1409.5 ± 48.13 AB	1346.17 ± 19.34 B	1376.67 ± 40.84 AB	1375.67 ± 26.10
	0.8 mm	1333.83 ± 18.49 B	1441.5 ± 62.79 A	1349.83 ± 47.00 B	1374.67 ± 35.89 AB	1374.96 ± 47.44
	1.2 mm	1353.83 ± 20.91 B	1364.5 ± 37.68 B	1369.83 ± 35.13 AB	1354 ± 35.94 B	1360.54 ± 7.95
Fracture toughness *K_Ic_* [MPa√m]						
		L1 (Mean ± SD)	L2 (Mean ± SD)	L3 (Mean ± SD)	L4 (Mean ± SD)	All Layers (Mean ± SD)
HTML	0.4 mm	4.27 ± 0.47 A	4.48 ± 0.60 A	4.42 ± 0.48 A	4.36 ± 0.62 A	4.38 ± 0.09
	0.8 mm	4.32 ± 0.32 A	4.73 ± 0.59 A	4.74 ± 0.97 A	4.53 ± 0.61 A	4.58 ± 0.20
	1.2 mm	4.98 ± 0.80 A	4.61 ± 0.68 A	4.87 ± 0.80 A	4.85 ± 0.81 A	4.83 ± 0.16
STML	0.4 mm	3.41 ± 0.59 A	3.26 ± 0.48 A	3.18 ± 0.46 A	3.05 ± 0.27 A	3.23 ± 0.15
	0.8 mm	3.83 ± 0.47 A	3.38 ± 0.50 A	3.09 ± 0.11 A	3.16 ± 0.83 A	3.37 ± 0.33
	1.2 mm	3.19 ± 0.51 A	3.50 ± 0.38 A	3.39 ± 0.64 A	3.35 ± 0.61 A	3.36 ± 0.13
UTML	0.4 mm	2.58 ± 0.67 A	2.90 ± 0.81 A	2.75 ± 0.48 A	2.66 ± 0.29 A	2.72 ± 0.14
	0.8 mm	2.61 ± 0.36 A	2.74 ± 0.29 A	2.54 ± 0.32 A	2.52 ± 0.27 A	2.60 ± 0.10
	1.2 mm	2.66 ± 0.31 A	2.94 ± 0.33 A	2.34 ± 0.31 A	2.77 ± 0.34 A	2.68 ± 0.25

Means that do not share the same letter (A,B,C,D,E) are significantly different, HTML: Katana™ Multi-layered zirconia; STML: Katana™ Super Translucent Multi-layered zirconia; UTML: Katana™ Ultra Translucent Multi-layered zirconia.

**Table 3 materials-16-00276-t003:** Descriptive non-parametric statistics (mean ± SD) for Weibull modulus *m* and characteristic strength *σ^w^* for all tested groups.

Weibull Modulus *m*						
		L1	L2	L3	L4	All layers
HTML	0.4 mm	5.27	7.53	9.35	8.48	6.02
	0.8 mm	9.18	11.95	8.67	6.52	8.54
	1.2 mm	9.36	6.42	9.95	8.70	8.96
STML	0.4 mm	5.22	2.60	2.12	3.63	5.84
	0.8 mm	7.43	6.71	3.88	3.73	6.07
	1.2 mm	7.77	6.15	6.97	5.43	6.03
UTML	0.4 mm	3.81	4.63	3.41	2.84	5.71
	0.8 mm	5.15	6.88	7.71	8.13	5.73
	1.2 mm	9.21	14.90	7.00	8.89	5.76
Characteristic Weibull strength *σ^w^* [MPa]						
		L1	L2	L3	L4	All layers
HTML	0.4 mm	708.76	813.95	813.24	806.14	790.93
	0.8 mm	727.39	756.97	764.95	809.97	757.08
	1.2 mm	705.52	907.26	835.06	938.73	837.41
STML	0.4 mm	336.55	364.39	364.26	311.71	344.04
	0.8 mm	379.92	467.00	420.42	415.66	433.16
	1.2 mm	384.17	456.80	455.64	363.58	415.37
UTML	0.4 mm	319.54	300.80	295.47	290.00	300.75
	0.8 mm	310.91	297.26	337.71	292.32	309.06
	1.2 mm	314.20	290.11	354.06	308.69	318.50

## Data Availability

Not applicable.

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
