# Peer review of "Mechanical Properties of New Generations of Monolithic, Multi-Layered Zirconia"

_materials, 2022, doi:10.3390/ma16010276_

Round 1

Reviewer 1 Report

1. Need to check the correspondance between the statistical significance and p value in the result section. 

Reviewer 2 Report

Dear authors,

the topic original or relevant in the field. It does address a specific gap in the field, especially because monolithic materials are gaining popularity.

The manuscript is well written, and the research is well performed.

I suggest (see below) supporting the discussion with some reference to cyclic fatigue simulation or in-vivo reports for these materials.

Nevertheless, here are some minor suggestions to improve your manuscript.

Table 1:

Translucency is generally reported as a number (TP, translucency parameter) or Contrast ratio ranging from 0 to 1. In the table, a percentage is used. Maybe is the light transmission? Please specify.

Line 265:

The authors wrote:

“There are variations in the quality of the production”

The sentence could be modified as follows:

“There are variations in the production process that may result in lots with different characteristics”

Lines 318-22:

The authors could add a sentence/paragraph outlining differences among monolithic materials available for restoration of posterior teeth outlining that differences have been reported even in single crown units submitted to cyclic fatigue:

Baldi A, Comba A, Ferrero G, Italia E, Michelotto Tempesta R, Paolone G, Mazzoni A, Breschi L, Scotti N. External gap progression after cyclic fatigue of adhesive overlays and crowns made with high translucency zirconia or lithium silicate. J Esthet Restor Dent. 2022 Apr;34(3):557-564. doi: 10.1111/jerd.12837. Epub 2021 Nov 16. PMID: 34783440; PMCID: PMC9298883.

In this study STML has been compared with LiSi showing higher external marginal gap progression. This could add a complete overview of the tested materials/characteristics to the paper.

From the results, the tables, and the charts, it appears that STML and UTML have close properties and that HTML shows marked differences. Especially for fracture toughness and flexural strength. The authors already discussed some discrepancies between reported data and manufacturer’s data, but maybe this “not so big” difference between STML and UTML could be outlined.

Please also add possible future research on this topic after considering your work's results.

Reviewer 3 Report

The study seems very interesting and may presents high value to the current dental literature, however the authors should address the following points to improve the quality of the manuscript:

- The abstract should be non-structured with specific word limits (please see the authors' guidelines)

- In the abstract, please add a short statement on the current gap in the dental literature to show the true value of this study.

- The aim of the study was mentioned clearly, however the authors should add the null (research) hypothesis/hypotheses at the end of the introduction sections.

- Please add a table in the materials and methods section to summarize the materials brands, manufacturers location and LOT numbers.

- The specimens look more like rectangular shape that bars, please change that in the manuscript.

- Usually the milling process for full or partial coverage restorations or prosthesis involve several layers of the same block/disc. Why did the authors consider splitting the blocks into layers? Is this process applicable for real scenarios?

- Why did the authors perform non-parametric tests for analysis. The analysis section seems very short. Please elaborate more for better clarity.

- Table 2 should be split into sections for better presentation.

- Figure 5 should be moved away from discussion section.

- Limitations and directions for future research should be added to the discussion section.

- Please consider summarizing the conclusion into concise bullet points.
